# The L467F-F508del Complex Allele Hampers Pharmacological Rescue of Mutant CFTR by Elexacaftor/Tezacaftor/Ivacaftor in Cystic Fibrosis Patients: The Value of the Ex Vivo Nasal Epithelial Model to Address Non-Responders to CFTR-Modulating Drugs

**DOI:** 10.3390/ijms23063175

**Published:** 2022-03-15

**Authors:** Elvira Sondo, Federico Cresta, Cristina Pastorino, Valeria Tomati, Valeria Capurro, Emanuela Pesce, Mariateresa Lena, Michele Iacomino, Ave Maria Baffico, Domenico Coviello, Tiziano Bandiera, Federico Zara, Luis J. V. Galietta, Renata Bocciardi, Carlo Castellani, Nicoletta Pedemonte

**Affiliations:** 1UOC Genetica Medica, IRCCS Istituto Giannina Gaslini, 16147 Genova, Italy; elvirasondo@gaslini.org (E.S.); valeriatomati@gaslini.org (V.T.); valeriacapurro@gaslini.org (V.C.); emanuela.pesce@yahoo.it (E.P.); mariateresalena@gaslini.org (M.L.); micheleiacomino@gaslini.org (M.I.); federico.zara@unige.it (F.Z.); renata.bocciardi@unige.it (R.B.); 2UOSD Centro Fibrosi Cistica, IRCCS Istituto Giannina Gaslini, 16147 Genova, Italy; federicocresta@gaslini.org; 3Department of Neurosciences, Rehabilitation, Ophthalmology, Genetics, Maternal and Child Health (DINOGMI), University of Genoa, 16132 Genova, Italy; cristinapastorino22@gmail.com; 4UOC Laboratorio di Genetica Umana, IRCCS Istituto Giannina Gaslini, 16147 Genova, Italy; avemariabaffico@gaslini.org (A.M.B.); domenicocoviello@gaslini.org (D.C.); 5D3-PharmaChemistry, Fondazione Istituto Italiano di Tecnologia, 16163 Genova, Italy; tiziano.bandiera@iit.it; 6Telethon Institute of Genetics and Medicine (TIGEM), 80078 Pozzuoli, Italy; l.galietta@tigem.it; 7Department of Translational Medical Sciences (DISMET), University of Naples Federico II, 80131 Naples, Italy

**Keywords:** correctors, potentiators, modulators, theratype, chloride secretion

## Abstract

Loss-of-function mutations of the *CFTR* gene cause cystic fibrosis (CF) through a variety of molecular mechanisms involving altered expression, trafficking, and/or activity of the CFTR chloride channel. The most frequent mutation among CF patients, F508del, causes multiple defects that can be, however, overcome by a combination of three pharmacological agents that improve CFTR channel trafficking and gating, namely, elexacaftor, tezacaftor, and ivacaftor. This study was prompted by the evidence of two CF patients, compound heterozygous for F508del and a minimal function variant, who failed to obtain any beneficial effects following treatment with the triple drug combination. Functional studies on nasal epithelia generated in vitro from these patients confirmed the lack of response to pharmacological treatment. Molecular characterization highlighted the presence of an additional amino acid substitution, L467F, in cis with the F508del variant, demonstrating that both patients were carriers of a complex allele. Functional and biochemical assays in heterologous expression systems demonstrated that the double mutant L467F-F508del has a severely reduced activity, with negligible rescue by CFTR modulators. While further studies are needed to investigate the actual prevalence of the L467F-F508del allele, our results suggest that this complex allele should be taken into consideration as plausible cause in CF patients not responding to CFTR modulators.

## 1. Introduction

Cystic fibrosis (CF) is the most frequent, life-threatening autosomal recessive disease, with an incidence, on average, between 1/3000 and 1/6000 in populations of European descent [1]. CF is a multiorgan condition affecting the upper and lower airways, the liver, the gastrointestinal and reproductive tracts, and the endocrine system, although morbidity and mortality are essentially due to the lung disease [2]. CF is caused by mutations in the gene encoding for the cystic fibrosis transmembrane conductance regulator (CFTR) anion channel. In the airways, loss of CFTR function at the apical membrane of epithelial cells results in defective chloride and bicarbonate secretion and, in parallel, to increased sodium absorption, leading to airway surface dehydration and impaired mucociliary clearance [2], which in turn result in a cycle sustaining infection and inflammation [2].

The CFTR channel is a member of the ATP-binding cassette (ABC) transporter superfamily [3,4], and is composed of two membrane-spanning domains (MSD1 and MSD2), two nucleotide-binding domains (NBD1 and NBD2), and a regulatory domain [3,4]. More than 2000 variants have been reported in the *CFTR* sequence, of which many affect the expression and/or function of the CFTR protein. According to the Clinical and Functional Translation of CFTR (CFTR2) database, to date, 382 sequence variants have been clearly defined as CF-causing (http://www.cftr2.org/index.php, accessed on 15 February 2022). *CFTR* mutations have been recently grouped into seven classes according to the type of defect they cause [5]. In particular, the class II group mutations cause misfolding of the nascent protein, resulting in its premature degradation, as in the case of F508del, which is the most frequent CF mutation [5]. Mutations grouped in class III determine a severely reduced activity due to a decreased open probability of the channel (gating defect; [5]). However, this classification of CF variants is complicated by the fact that many mutations display more than one type of defect, and thus can be assigned to more than one class [6]. This is the case of F508del, which, in addition to the class II trafficking defect, also displays a gating impairment, although not as severe as in other class III mutations. For this reason, it has been proposed to categorize pathogenic variants not only according to their functional impact on the CFTR protein but also to the protein response to CFTR modulators; i.e., to their theratype [7]. Modulators are compounds that can partially correct the CFTR defects and rescue protein function [8]. They include correctors, which are small molecules able to rescue the folding defect of class II mutants. Lumacaftor (Luma; [9]), tezacaftor (Teza; [10]), and elexacaftor (Elexa; [11]) are all correctors. A combination of elexacaftor with either lumacaftor or tezacaftor results in a synergistic effect with improved rescue of CFTR misfolding [12,13]. Gating defects can be ameliorated by other compounds called potentiators, such as ivacaftor (Iva; [14]). Combinations of modulators have been developed to rescue CFTR mutants displaying more than one type of defect [8,12].

Some people with CF carry complex alleles, with more than one CFTR mutation on the same allele (i.e., in cis). In some cases, the molecular and functional defect of the variant protein might be different from that originated by the individual mutations combined in the complex allele, as each variant can affect different steps in CFTR biogenesis, often with unpredictable cumulative effects. Conversely, some variants could be neutral or even have a beneficial role, such as the so-called “revertant mutations”, which mitigate the functional defect caused by the mutation in cis [15]. Thus, complex alleles could be associated with a response to CFTR modulators different from that of the variants singularly considered [16]. Importantly, when a common mutation is identified by a standard molecular panel, other possible in cis variants are not routinely searched, making their frequency probably underestimated [16]. This happens also with complex alleles known to be particularly frequent in certain regional cohorts, such as A238V-F508del and F508del-I1027T [16]. Next-generation sequencing (NGS) techniques allow to investigate the complete *CFTR* locus in CF patients, leading to the identification of complex alleles that could hamper responsiveness to modulator-based therapies.

The development of novel predictive models based on the use of patients’ intestinal [17] or nasal epithelial (HNE) cells [18,19,20,21,22] has allowed to characterize—in a native system—the molecular and functional protein defect caused by variants and to determine the response to current therapies. Several studies have demonstrated the value of HNE cells as a disease-relevant, ex vivo model [18,19,20,21,22,23].

Recently, a new drug composed of the CFTR modulators elexacaftor, tezacaftor, and ivacaftor (ETI to indicate the triple combination) has achieved a breakthrough in the management of CF disease in patients carrying one or two F508del alleles [24,25]. Although the gain in lung function and improvement in other critical parameters is on average of an extent never seen before in CF, there is a marked variability in the clinical response, with few individuals faring rather poorly. At least a fraction of these cases could be related to undetected complex alleles. This can be investigated by the HNE ex vivo model through its potential to predict response to modulators.

In this work, using ex vivo HNE and heterologous expression systems, we employed an extensive molecular and functional analysis to establish the mechanism of the lack of response to ETI treatment in two F508del compound heterozygotes, and prove it to be an *in cis* mutation that abolishes the response of F508del to ETI.

## 2. Results

This study stemmed from the observation of a CF patient (ID: TT001), compound heterozygous for the F508del and G542X CFTR variants, and therefore considered eligible for ETI administration, who failed to obtain any beneficial effects following the treatment with the drug, as expected for his genotype (clinical data of patient TT001 can be found in Appendix A).

In an attempt to understand the reason(s) for the lack of response to ETI, we decided to directly evaluate the CFTR activity and pharmacological rescue in the patient’s nasal cells by means of electrophysiological techniques.

Nasal cells derived from nasal brushing of both nostrils of the patient (TT001) were cultured as described in the Methods, Section 4.2, plated onto permeable supports, and differentiated into nasal epithelia under air-liquid conditions. Epithelia were then treated for 24 h by adding the compounds to the basolateral medium. The following day, the nasal epithelia were mounted in a perfusion chamber to perform short-circuit current measurements and thus to determine CFTR-dependent Cl^−^ secretion (Figure 1A). The epithelial sodium channel ENaC was first inhibited by adding amiloride (10 µM), and then the cells were stimulated with CPT-cAMP (a membrane permeable cAMP analogue; 100 µM) followed by ivacaftor (1 µM) to maximally activate F508del-CFTR. Finally, the CFTR currents were blocked by adding the CFTR inhibitor-172 (inh-72, 20 µM) (Figure 1A). For each epithelium, the total CFTR activity was estimated as the amplitude of the current drop caused by the addition of inh-172.

Total CFTR activity recorded in negative control epithelia (i.e., epithelia treated with vehicle alone, 0.18% DMSO) was very small: 1.6 ± 0.4 µA (mean ± SD, *n* = 4; Figure 1A,B). We then evaluated the CFTR activity in epithelia treated only with correctors (either elexacaftor alone, or combined Elexa/Teza), thus in the absence of chronic ivacaftor, or with ETI. Indeed, various studies have shown that the potentiator ivacaftor, upon long-term treatments, impairs the rescue of F508del-CFTR by correctors. It has been proposed that this is due to a destabilization of the protein, increasing its turnover rate [26,27]. This effect has been demonstrated following rescue by lumacaftor [26,27] and, more recently, also by Elexa/Teza combination [28]. We thus postulated that the detrimental effect of ivacaftor could be particularly strong in the TT001 patient. Disappointingly, we could not detect any increase in the extent of CFTR-mediated current neither in epithelia treated with correctors Elexa/Teza alone nor in epithelia treated with ETI (Figure 1A,B). For comparison, we performed similar analyses on nasal epithelia derived from a F508del homozygous patient (TT003) who is receiving the triple combination therapy with a marked clinical benefit (clinical data of patient TT003 can be found in Appendix A). In DMSO-treated epithelia, CFTR activity was small (3.2 ± 0.4 µA; mean ± SD, *n* = 4); however, treatment with Elexa/Teza resulted in a significant four-fold increase in the CFTR-mediated current, as compared to vehicle-treated cells (Figure 1C,D). Interestingly, the same increase in CFTR activity was obtained in epithelia treated with ETI, demonstrating that, under these conditions, no apparent detrimental effect of chronic VX-770 treatment could be detected (Figure 1C,D). Thus, ex vivo testing of CFTR modulators on the TT001 patient’s cells confirmed the lack of response to the triple combination.

We therefore looked in more detail into the molecular characterization of the patient, through the analysis of CFTR cDNA obtained by reverse transcription of the total RNA extracted from the patient’s differentiated epithelia. We amplified and sequenced different RT-PCR products spanning the two variants F508del and G542X from exon 9 to 14 and we found that actually the patient was carrying a third substitution, the L467F previously unidentified. As shown in Figure 2A, cDNA sequence analysis allowed the detection of an expected allelic imbalance due to the mRNA degradation by the nonsense-mediated decay of the G542X carrying transcript and revealed that the F508del and L467F variants were present in cis on the same allele (Figure 2B). This was further confirmed by sequencing analysis after PCR product subcloning (not shown).

Concomitantly with the identification of the complex allele in patient TT001, another patient (TT190) who recently entered ETI treatment appeared not to respond to the drugs (clinical data of patient TT190 can be found in Appendix A). The referred genotype of this patient was F508del/E585X. However, prompted by our previous finding, the patient was immediately sent for an in-depth molecular screening, with segregation analysis, which identified the same complex allele, with the presence of the L467F and F508del variants in cis.

Analysis of CFTR-mediated chloride secretion in the patient’s derived nasal epithelia confirmed the lack of rescue by treatment with CFTR-modulating drugs, as previously observed for patient TT001 (Figure 3).

We investigated in vitro the pharmacological responsiveness to CFTR modulators of the L467F-F508del complex allele. To this aim, we expressed the variant protein, carrying both mutations, into two different cell models widely used for studies on CFTR pharmacology and biology, FRT and CFBE41o^−^ cells, both with stable expression of the halide-sensitive Yellow Fluorescent Protein (HS-YFP). As additional controls, we also expressed the variant protein resulting from the single mutations, i.e., F508del- and L467F-CFTR, as well as the wildtype CFTR expression constructs.

Under these experimental settings, F508del-CFTR showed a clear positive response to rescue agents, particularly to the corrector combination Elexa/Teza plus the potentiator ivacaftor (Figure 4). The triple combination resulted in an overall rescue of the CFTR activity, which was approximately 50% that measured in cells transfected with the wild-type form of the CFTR protein (Figure 4). The L467F-CFTR mutant displayed a marked residual activity upon cAMP increase (approximately equal to 60% of the activity of wild-type CFTR) that was further increased following treatment with CFTR modulators up to 80% of the activity of wt-CFTR (Figure 4). On the contrary, the protein produced by the complex allele (L467-F508del CFTR) displayed a severely reduced activity, with no or negligible rescue by CFTR modulators (Figure 4).

To confirm these results, we evaluated biochemically the expression of the CFTR protein in FRT and CFBE41o^−^ cells transiently transfected with F508del, L467F, or L467F-F508del CFTR. In both cell models, in cells transfected with F508del CFTR, and treated with the vehicle alone, the prevalent form was the immature, core-glycosylated CFTR protein. After treatment with combined Elexa/Teza, the mature, fully glycosylated CFTR form became clearly evident (Figure 5A,B). Cells transfected with L467F CFTR, instead, expressed both the immature and the mature forms of the CFTR protein. Treatment with Elexa/Teza markedly increased the expression of the mature form (Figure 4A,B). In agreement with the functional data, cells expressing L467F-F508del CFTR expressed mainly the immature form of the protein (Figure 5A,B). Treatment with Elexa/Teza resulted in very modest expression of the mature form of CFTR protein in CFBE41o^−^ cells, while in FRT cells the treatment was not effective (Figure 5A,B).

We therefore analyzed the CFTR expression pattern on nasal epithelia derived from patient TT001 under resting conditions and following treatment with CFTR modulators. As shown in Figure 5C, no mature CFTR protein expression was detected under any experimental condition, suggesting that, in native cells, the maturation defect is so severe to preclude expression of the mutant.

## 3. Discussion

CF represents a model disease in many respects, and its research history, more than in other disorders, has been marked by a close cooperation between scientists and clinicians. Since 1989 and the discovery of the causative gene and by reverse genetics of the CFTR protein [29], remarkable efforts have been made to identify compounds that could rescue the mutant CFTR expression and function. In the last ten years, four new drugs have been authorized for the treatment of CF patients with specific genotypes. The newest ETI combination (brand names Trikafta^®^ in US and Kaftrio^®^ in Europe, Vertex Pharmaceuticals, Boston, MA, USA) has demonstrated unprecedented positive outcomes in clinical trials [24,25] on patients bearing one or two copies of the F508del allele. Recently, its use has been extended also to patients bearing folding/trafficking CFTR variants other than F508del. Although modulators have undoubtedly proved to be a breakthrough in CF treatment, sporadic reports of non-responders among the patients under treatment with CFTR modulators have begun to emerge [24].

The presence of non-responders is not unusual in pharmacological therapies and can be ascribed to several factors [30]. In general, the lack of response to a drug results from a particular and patient-specific combination of genetic factors. The most common causes are genetic variants in enzymes involved in metabolism, typically polymorphism in one of the Cytochrome P450 enzymes, which may significantly alter the pharmacokinetics of the drug [31]. For instance, patients with CyP450 allelic variants that qualify them as “ultra-rapid metabolizers” may not benefit from treatment with specific drugs as they are very rapidly cleared from the circulation and do not reach the biological target in sufficient amount to elicit the pharmacological response. A precision medicine approach with pharmacogenetic testing would pinpoint the exact mechanism, but this may not be always feasible in clinical practice.

Other non-genetic factors affecting metabolism may result from inhibition or induction of metabolizing enzymes by drugs. Among the CFTR modulators, it is known that lumacaftor is an inducer of CyP450 3A enzymes, and that steady-state exposure of ivacaftor is reduced when administered in combination with lumacaftor [32,33]. In clinical settings, such an event can be assessed by Therapeutic Drug Monitoring (TDM), but unfortunately the measurement of plasma levels of modulators and their metabolites is far from being standard practice and is available only in very few laboratories. Implementation of TDM analyses in CF centers would be therefore highly desirable.

Poor response to therapies can be explained also by limited compliance to the prescribed medications, a well-known factor in unsatisfactory therapeutic results. The heavy burden of care of the average patient with CF concurs to explain the suboptimal adherence, which is also influenced by route of administration, age, season, and by the concomitant busy schedule of other daily activities [34]. Adherence to a highly effective modulator such as ivacaftor is higher than to other medicines used in CF, such as inhaled dornase alfa, hypertonic saline, oral pancreatic enzyme and vitamin supplements, and airway clearance therapy [34], but still reported to be only 61% in electronically monitored patients [35]. Sub-optimal compliance to modulators may be further deteriorated by the fact that, in order to be absorbed correctly, they need to be taken together with fat-containing food, which sometimes does not fit with the dietary routine of patients. However, in the cases described here, there was no evidence of poor adherence, and the poor clinical response to the ETI combination was observed also during prolonged hospitalization where the drug administration was monitored and optimal.

Importantly, in our in vitro studies on cultured nasal epithelial cells from the TT001 patient, we found no CFTR rescue following treatment with ETI combination. On the contrary, we detected a marked CFTR rescue on HNE derived from the TT003 patient, successfully treated with ETI. This implies that the lack of drug effect on the TT001 patient’s cells is due to an intrinsic defect that is maintained during ex vivo/in vitro testing, and not related to drug absorption, distribution, or metabolism in vivo.

Another plausible cause for poor drug response, the detrimental effect caused by chronic treatment with the potentiator ivacaftor, was considered [26,27]. However, in our in vitro experiments, we found no difference between epithelia treated for 24 h with elexacaftor/tezacaftor, and then with ivacaftor acutely (during recordings) or chronically for 24 h.

Finally, we hypothesized the presence of an undetected complex allele. A molecular characterization of the patients’ CFTR mRNA sequence was performed, disclosing a third substitution, L467F, located in cis with F508del. L467F-CFTR has been reported to be damaging to protein maturation, and the L467F-F508del complex allele not responsive to lumacaftor treatment [36]. This study proceeded in that direction by testing single or combined CFTR correctors on L467F-F508del CFTR and confirming the severe maturation defect. The immature, core-glycosylated CFTR protein expressed was not rescued by correctors, being not only insensitive to lumacaftor but also to the more effective elexacaftor/tezacaftor combination.

The presence of a complex allele does not always affect the responsiveness to CFTR modulators [36]. However, as we demonstrate in this report, it is fundamental to identify multiple mutations on the same allele as the overall effect in response to treatment with CFTR modulators may depend on the specific combination of variants and may require a case-by-case analysis. The F508del variant has a high frequency in the general population (1/25), while L467F is rare in the GnomAD database (MAF = 5.91 × 10^−5^). In our institute’s in-house cohort, the Gaslini exome genomic database built up of 1756 individuals with no family history of CF, we identified 27 (27/1756 = 1.5%) heterozygotes for the p.F508del variant, including 1 individual (1/27 = 3.5%) carrying p.F508del and p.L467F in cis (Appendix A). In spite of the relatively small cohort, this seems to suggest that this complex allele might be not so infrequent, and it should be taken into account when considering prescription of mutation-specific therapies.

As a matter of fact, the assessment of the actual impact of complex alleles in poor response to ETI treatment may prove challenging if only medical records are considered. A significant fraction of patients was genetically tested several years ago, often using limited panels, and the molecular diagnosis was concluded when two CFTR variants were identified. Hence, some patients could unknowingly carry complex alleles, which in turn may affect the functional mechanism of the protein and its responsiveness to modulators, as in the case presented here. It remains to be determined if this complex allele, whose actual prevalence is unknown and possibly ethnically variable, should be actively looked at for before prescribing CFTR modulators or, on the other hand, only in those cases where clinical and sweat test post-treatment results are unsatisfactory and alternative explanations lacking.

In any case, due to the not infrequent adverse events associated with the use of CFTR modulators and to their relevant economic impact on health systems, it remains important to identify non-responders to treatment and investigate the mechanisms behind the therapeutic failure. If no action can be implemented to achieve the desired therapeutic effect, such as in the case described here, treatment discontinuation should be discussed with the patient.

Further lines of research may follow this finding, including a consensus on the definition of a non-responder to CFTR modulators and on investigations to implement in these situations, a more precise estimation of their frequency, and studies on alternative complex alleles that might affect the efficacy of these compounds.

## 4. Materials and Methods

### 4.1. Patients under Study

Two patients compound heterozygous for F508del and a minimal function CFTR variant and non-responders to ETI treatment were considered (donor IDs: TT001 and TT190). A third CF patient, homozygous for F508del, was included as the control for ETI treatment (donor ID: TT003). Their clinical data are shown in Appendix A. As further controls, two healthy subjects were enrolled (Ctr01 and Ctr32).

### 4.2. Primary Nasal Epithelial Cell Culture

Isolation, culture, and differentiation of primary airway epithelial cells were performed as previously described [37], with some modifications. Nasal epithelial cells were obtained through a nasal brushing of both nostrils. Airway epithelial cells were cultured in a serum-free medium (LHC9 mixed with RPMI 1640, 1:1) containing various hormones and supplements, which favors cell number expansion, including ROCK and SMAD inhibitors (DMH-1, A-83-01, and Y-27632 compounds) [38]. The culture medium contained in the first days a mixture of different antibiotics (including colistin, piperacillin, and tazobactam) to eradicate bacterial contamination.

To obtain differentiated epithelia, nasal cells were seeded at high density (500,000 cells/cm^2^) on porous membranes (Snapwell inserts, code 3801, Corning Life Sciences, Corning, NY, USA). After 24 h, the serum-free medium was removed from both sides and, on the basolateral side only, replaced with Pneumacult ALI medium (StemCell Technologies, Vancouver, BC, Canada), and differentiation of cells (up to 16–18 days) was performed in air-liquid interface (ALI) condition.

### 4.3. Short-Circuit Current Recordings

Snapwell inserts carrying differentiated nasal epithelia were mounted in a vertical diffusion chamber resembling a Ussing chamber with internal fluid circulation. Both apical and basolateral hemichambers were filled with 5 mL of a solution containing (in mM) 126 NaCl, 0.38 KH_2_PO_4_, 2.13 K_2_HPO_4_, 1 MgSO_4_, 1 CaCl_2_, 24 NaHCO_3_, and 10 glucose. Both sides were continuously bubbled with a gas mixture containing 5% CO_2_–95% air and the temperature of the solution was kept at 37 °C. The transepithelial voltage was short-circuited with a voltage-clamp (DVC-1000, World Precision Instruments, Sarasota, FL, USA; VCC MC8 Physiologic Instruments, Reno, NV, USA) connected to the apical and basolateral chambers via Ag/AgCl electrodes and agar bridges (1 M KCl in 1% agar). The offset between the voltage electrodes and fluid resistance was adjusted to compensate for the parameters before experiments. The short-circuit current was recorded by analogical to digital conversion on a personal computer.

### 4.4. RNA Extraction and cDNA Analysis

Patients’ cells from differentiated nasal epithelia were collected from permeable supports by scraping, and total RNA was extracted by using the TRIzol reagent (Ambion-ThermoFisher Scientific, Waltham, MA, USA) by a standard protocol followed by a clean-up procedure with the Rneasy Mini kit (Qiagen, Germantown, MD, USA) as suggested by the manufacturer. RNA was checked and quantified with a Nanodrop Spectrophotometer (ThermoFisher Scientific, Waltham, MA, USA), and first-strand cDNA was synthesized by the High-Capacity cDNA Reverse Transcription Kit (Applied Biosystems-ThermoFisher Scientific, Waltham, MA, USA) from 200 ng of total RNA. Qualitative analysis of the CFTR transcript was performed by PCR amplification with specific oligonucleotides (sequences available upon request) spanning the cDNA region bearing the variants (rs1800089, NM_000492.4:c.1399C>T; NP_000483.3:p.Leu467Phe; rs113993960, NM_000492.4:c.1521_1523del; NP_000483.3:p.Phe508del and rs113993959, NM_000492.4:c.1624G>T; NP_000483.3:p.Gly542Ter, nomenclature according to the *HGVS* guidelines). Briefly, 4 microL of cDNA diluted 1:2 were used as a template to amplify CFTR mRNA with GoTaq G2 Polymerase mastermix (Promega, Madison, WI, USA). PCR products were checked by agarose gel electrophoresis, purified by enzymatic digestion with EXO-SAP Exo/SAP-IT (ThermoFisher Scientific, Waltham, MA, USA) and then used for direct sequencing. Reactions were set up with a Big Dye Terminator Cycle Sequencing Kit as suggested by the provided protocol, run on a 3130xl Genetic Analyzer (Applied Biosystems-ThermoFisher Scientific, Waltham, MA, USA) and sequences analyzed by the Sequencer 4.7 software.

### 4.5. Genomic Analysis

Peripheral blood samples were collected upon informed consent administration. Genomic DNA was extracted by a standard, automated procedure (QIA Symphony QIAGEN). Genomic DNA fragments corresponding to *CFTR* exons 11 and 12 were amplified by PCR with specific primers (sequences available upon request) using Platinum Taq DNA polymerase (Invitrogen-ThermoFisher Scientific, Waltham, MA, USA). The PCR fragments were directly sequenced in both strands using a Big-Dye-Terminator v1.1 Cycle Sequencing kit (Applied Biosystems-ThermoFisher Scientific, Waltham, MA, USA). Sequencing reactions were purified by automated Agencourt CLEANSEQ System (Beckman Coulter, Brea, CA, USA) and run and analysed on a 3730xl DNA Analyzer (Applied Biosystems-ThermoFisher Scientific, Waltham, MA, USA).

### 4.6. Cell Culture

FRT and CFBE41o^−^ cells stably expressing the halide-sensitive yellow fluorescent protein (HS-YFP) YFP-H148Q/I152L and F508del-CFTR were generated as previously described [39,40]. FRT cells were grown in Coon’s F-12 modified medium while CFBE41o^−^ required MEM, in both cases supplemented with 10% fetal calf serum, 2 mM L-glutamine, 100 U/mL penicillin, and 100 µg/mL streptomycin. For the YFP-based assays of CFTR activity, FRT or CFBE41o^−^ cells were plated (100,000 cell/well or 50,000 cells/well, respectively) on clear-bottom 96-well black microplates (Corning Life Sciences, Corning, NY, USA).

### 4.7. Antibodies, Vectors, and Chemicals

The following antibodies were used: mouse monoclonal anti-CFTR (ab769 and ab596, J.R. Riordan, University of North Carolina at Chapel Hill, and Cystic Fibrosis Foundation Therapeutics); mouse monoclonal anti-GAPDH (sc-32233; Santa Cruz Biotechnology, Inc.; RRID:AB_627679); horseradish peroxidase (HRP)-conjugated anti-mouse IgG (ab97023; Abcam; RRID:AB_10679675).

Vectors encoding wt-, L467F-, F508del-, and L467F-F508del-CFTR variants were purchased from VectorBuilder (vector IDs available upon request; Neu-Isenburg, Germany).

The CFTR modulators ivacaftor, tezacaftor, and lumacaftor were from TargetMol (catalog ID: T2588, T2263, and T2595, respectively; Wellesley Hills, MA, USA) while elexacaftor was purchased from MedChemExpress (catalog ID: HY-111772; Monmouth Junction, NJ, USA). The final working concentration used for the CFTR modulators were as follows: elexacaftor, 3 µM; tezacaftor, 10 µM; lumacaftor, 3 µM; ivacaftor, 1 µM (when applied acutely during short-circuit current measurements or for the YFP assay) or 5 µM (for 24 h treatment in the ETI combination).

### 4.8. Transient Transfection of FRT and CFBE41o^−^ Cell Lines

To perform the microfluorimetric YFP-based assay, FRT and CFBE41o^−^ cells expressing the HS-YFP were reverse-transfected onto 96-well plates with 0.2 μg per well of the indicated vectors (see dedicated Methods Section 4.7). To measure CFTR expression by Western blot, cells were reverse-transfected onto 6-well plates with 2 µg of the indicated vectors (see dedicated Methods Section 4.7). Lipofectamine 2000 (ThermoFisher Scientific, Waltham, MA, USA) was used as a transfection agent. Cells were transfected, in Opti-MEM™ Reduced Serum Medium (ThermoFisher Scientific, Waltham, MA, USA). After 6 h, Opti-MEM was carefully replaced with culture medium without antibiotics. A total of 24 h after transfection and plating, cells were treated with correctors or vehicle alone (DMSO) at the indicated concentrations and incubated at 37 °C for an additional 24 h, prior to proceeding with the functional HS-YFP-based assay or to the cell lysis.

### 4.9. YFP-Based Assay for CFTR Activity

CFTR activity was determined by the HS-YFP microfluorimetric assay, described in detail in previous studies [41,42,43]. Briefly, prior to the assay, cells were washed with PBS (137 mM NaCl, 2.7 mM KCl, 8.1 mM Na_2_HPO_4_, 1.5 mM KH_2_PO_4_, 1 mM CaCl_2_, and 0.5 mM MgCl_2_) and then incubated for 25 min with 60 µL of PBS plus forskolin (20 µM) and VX-770 (1 µM) to maximally stimulate the CFTR channel. For CFTR activity determination, cells were then transferred to a microplate reader (FluoStar Galaxy or Fluostar Optima; BMG Labtech, Offenburg, Germany), equipped with high-quality excitation (HQ500/20X: 500 ± 10 nm) and emission (HQ535/30M: 535 ± 15 nm) filters for YFP (Chroma Technology, Bellows Falls, VT, USA). Each assay consisted of a continuous 14-s YFP fluorescence recording with 2 s before and 12 s after injection of 165 µL of an iodide-containing solution (PBS with Cl^−^ replaced by I^−^; final I^−^ concentration 100 mM). Data were normalized to the initial background-subtracted fluorescence. To determine the I^−^ influx rate, the final 11 s of the data for each well were fitted with an exponential function to extrapolate the initial slope (dF/dt).

### 4.10. Western Blot

To generate the primary HNE cells’ lysates the following protocol was applied. In order to remove mucus excess, the apical side of HNEC epithelia differentiated in ALI conditions for 16 days were washed with warm HBSS containing 0.4% sodium bicarbonate for 3 h at 37 °C. Cells were then washed twice with warm complete PBS. After washing, the apical side of the filters was dried while basolateral culture medium (Pneumacult ALI; StemCell Technologies, Vancouver, BC, Canada) was changed to treat cells with indicated correctors or vehicle (DMSO) for 24 h at 37 °C.

The following day, prior to cell lysis, HNE cells were washed again once with warm HBSS 0.4% sodium bicarbonate at 37 °C for 30 min to remove the newly produced mucus. Cells were washed once with warm complete PBS and three times with ice-cold PBS without calcium and magnesium plus proteases inhibitors cocktail (Roche, Rotkreuz, Switzerland) on ice. Epithelia were lysed on ice by applying 100 µL/filter of ice-cold RIPA buffer plus proteases inhibitors (Roche, Rotkreuz, Switzerland). Cell layers were scraped, collected in a tube, and left on ice for 15 min. To reduce the lysate viscosity, 5 × 22 G needle syringe passages followed by 5 × 27 G needle syringe passages were applied. Lysates were then cleared by centrifugation (15,000× *g* for 20 min at 4 °C). After centrifugation, the supernatant was transferred to a new tube and stored at −80 °C for subsequent analysis.

After transfection, FRT and CFBE41o^−^ cells were grown to confluence onto a 6-well plate. The day of cell lysis, cells were washed with ice-cold D-PBS without Ca^2+^/Mg^2+^ and then lysed in RIPA buffer (50 mM Tris-HCl pH 7.4, 150 mM NaCl, 1% Triton X-100, 0.5% Sodium deoxycholate, 0.1% SDS) containing a complete protease inhibitor cocktail (Roche, Rotkreuz, Switzerland). Cell lysates were then processed as previously described [41]. In brief, lysates were separated by centrifugation at 15,000× *g* at 4 °C for 10 min.

The supernatant protein concentration was calculated using a BCA assay (ThermoFisher Scientific, Waltham, MA, USA) following the manufacturer’s instructions. Proteins (25 µg for cell lines and 50 µg for HNE cells epithelia) were separated onto gradient 4–15% Criterion TGX Precast gels (Bio-rad Laboratories Inc., Hercules, CA, USA), transferred to a nitrocellulose membrane with a Trans-Blot Turbo system (Bio-rad Laboratories Inc., Hercules, CA, USA) and analyzed by Western blotting. CFTR and GAPDH were detected using antibodies indicated in the dedicated Methods Section 4.7 and subsequently visualized by chemiluminescence using the SuperSignal West Femto or West Dura Substrate (ThermoFisher Scientific, Waltham, MA, USA). Molecular Imager ChemiDoc XRS System (Bio-rad Laboratories Inc., Hercules, CA, USA) was used to monitor the chemiluminescence. Images were analyzed with ImageJ software (National Institutes of Health, Bethesda, MD, USA). Bands were analyzed as region-of-interest (ROI), normalized against the GAPDH loading control.

### 4.11. Statistics

The Kolmogorov–Smirnov test was used to assess the assumption of normality of data distribution. An analysis of variance (ANOVA) followed by a post-hoc test was used when comparing more than two groups in order to avoid “multiple comparisons error”. For normally distributed quantitative variables, a parametric ANOVA was performed.

Statistical significance of the effect of single drug treatments on CFTR activity in FRT, CFBE41o^−^, or HNE cells was tested by parametric ANOVA followed by the Dunnet multiple comparisons test (all groups against the control group) as a post-hoc test. In the case of combinations of drugs, statistical significance was verified by ANOVA followed by the Tukey test (for multiple comparisons) as a post-hoc test. When comparing selected pairs of treatment, the statistical significance was tested by ANOVA followed by Bonferroni as a post-hoc test.

Normally distributed data are expressed as the mean ± SD and significances are two-sided. Differences were considered statistically significant when *p* < 0.05.

## Figures and Tables

**Figure 1 ijms-23-03175-f001:**
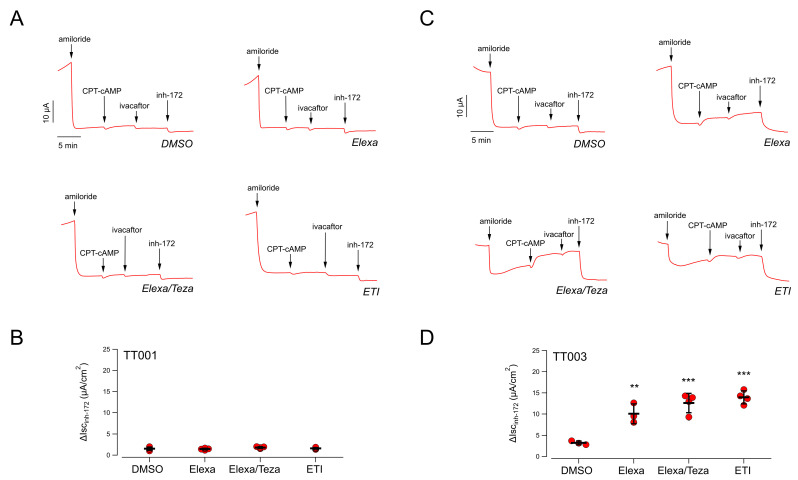
Functional evaluation of the elexacaftor-based combinations on nasal epithelia derived from CF patients bearing the F508del mutation. (**A**) Representative traces of the effect of the vehicle alone (DMSO) or Elexa (3 µM), or Elexa/Teza (3 µM/10 µM) or ETI (3 µM/10 µM/5 µM) in F508del/G542X nasal epithelial cells (derived from donor ID: TT001) with the short-circuit current technique. (**B**) Scatter dot plot showing the summary of results obtained from experiments described in (**A**) Data reported are the amplitude of the current blocked by 20 µM inh-172 (ΔIsc_inh-172_). (**C**) Representative traces of the effect of treatment with modulators (same conditions tested in (**A**)) on F508del/F508del nasal epithelial cells (derived from donor ID: TT003) using the short-circuit current technique. (**D**) Scatter dot plot showing the summary of ΔIsc_inh-172 data_ obtained from experiments described in (**C**). Asterisks indicate statistical significance of the treatments vs. control (DMSO-treated): ** *p* < 0.01; *** *p* < 0.001.

**Figure 2 ijms-23-03175-f002:**
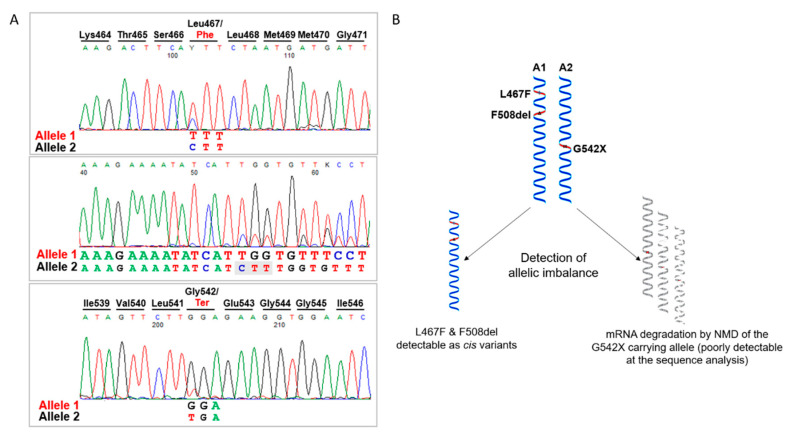
cDNA analysis and haplotype reconstruction. (**A**) *CFTR* cDNA spanning the regions containing the variants was amplified by RT-PCR and the obtained products checked by Sanger sequencing, thus disclosing the presence of a third substitution, the L467F. As shown, we detected an allelic imbalance due to NMD mRNA degradation of the G542X-carrying transcript with the corresponding sequence peaks less intense compared to those showing the presence of the L467F and F508del variants, thus indicating that these latter are in cis and configure a complex allele. (**B**) Schematic representation of the allelic phase definition. The complex allele is also carrying the common V470M polymorphism frequently observed associated with pathogenic CFTR variants. A1, allele 1; A2, allele 2; NMD, nonsense-mediated mRNA decay.

**Figure 3 ijms-23-03175-f003:**
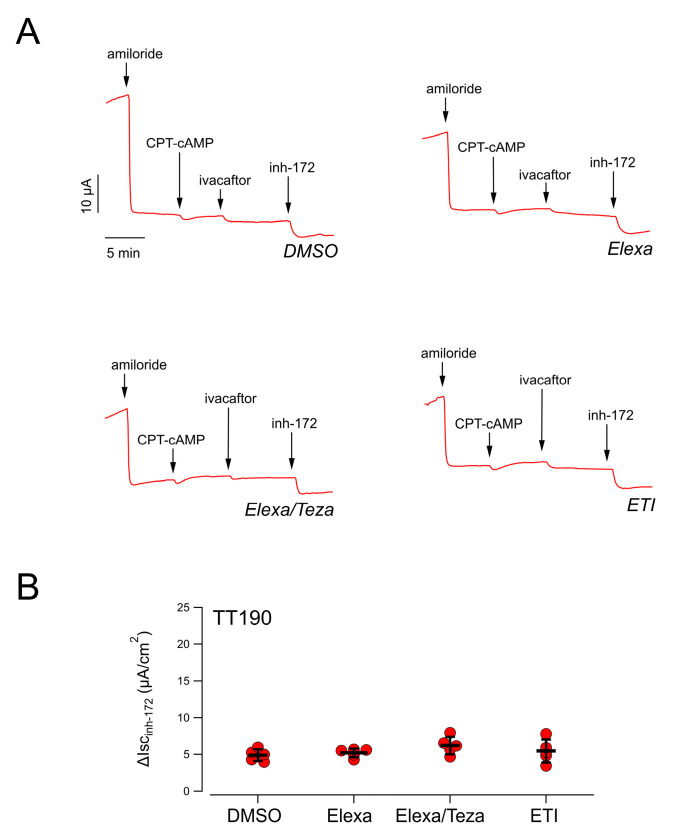
Functional evaluation of elexacaftor-based combinations on nasal epithelia derived from the TT190 patient (L467F-F508del/E585X). (**A**) Representative short-circuit current recordings on epithelia derived from patient’s nasal cells treated for 24 h with vehicle alone (DMSO) or Elexa (3 µM), or Elexa/Teza (3 µM/10 µM) or ETI (3 µM/10 µM/5 µM). (**B**) Scatter dot plot showing the summary of results obtained from experiments described in (**A**). Data reported are the amplitude of the current blocked by 20 µM inh-172 (ΔIsc_inh-172_).

**Figure 4 ijms-23-03175-f004:**
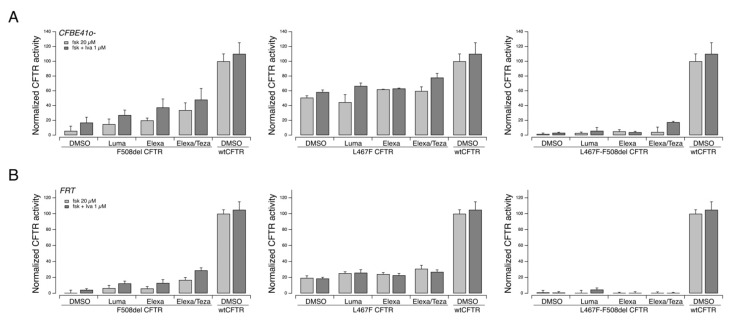
Functional evaluation of the activity and response to pharmacological treatments of the L467F-F508del-CFTR complex allele on heterologous expression systems. The bar graphs show the activity of L467F-F508del-CFTR and, for comparison, of F508del- and L467F-CFTR transiently expressed in CFBE41o^−^ (**A**) or FRT (**B**) cells stably expressing the HS-YFP. CFTR activity was determined as a function of the YFP quenching rate following iodide influx in cells treated for 24 h with DMSO alone (vehicle), or Luma (3 µM), or Elexa (3 µM), or Elexa/Teza (3 µM/10 µM).

**Figure 5 ijms-23-03175-f005:**
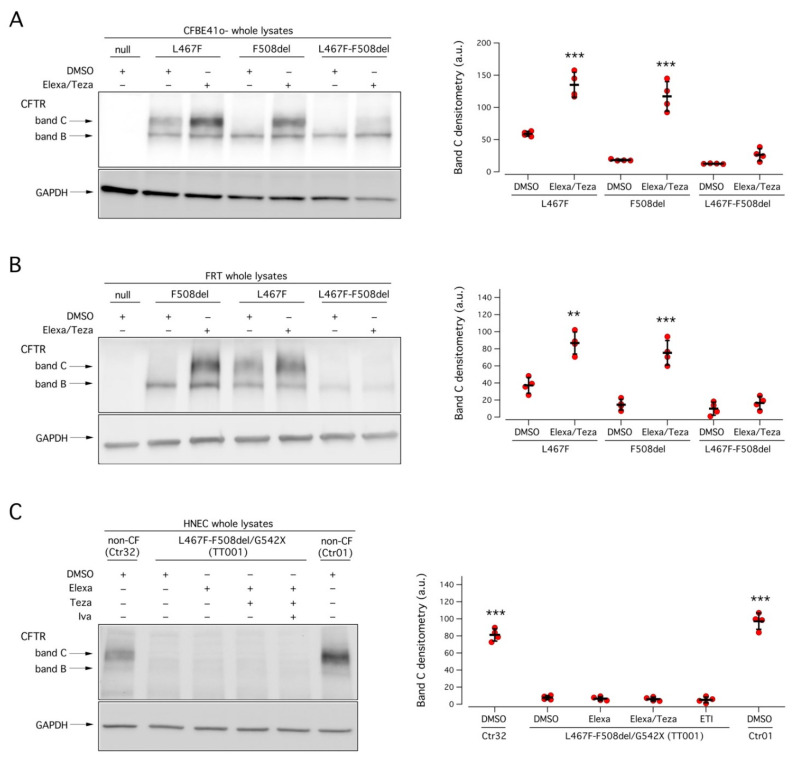
Biochemical analysis of the L467F-F508del CFTR complex allele expression pattern. (**A**,**B**) Left panels: representative Western blot images showing the electrophoretic mobility of L467F-F508del-CFTR and, for comparison, of F508del- and L467F-CFTR transiently expressed in CFBE41o^−^ (**A**) or FRT (**B**), treated for 24 h with DMSO alone (vehicle), or Elexa/Teza (3 µM/10 µM). Arrows indicate the complex-glycosylated (band **C**) and core-glycosylated (band **B**) forms of the CFTR protein. Right panels: CFTR band (**C**) densitometry of the Western blot experiments. Asterisks indicate statistical significance of the treatments vs. their respective control (DMSO-treated): ** *p* < 0.01; *** *p* < 0.001. (**C**) Left panel: representative Western blot image showing CFTR electrophoretic mobility in lysates of nasal epithelia derived from patient TT001, and, for comparison, from two non-CF donors (ID: Ctr01 and Ctr32). Epithelia were treated for 24 h with vehicle alone (DMSO) or Elexa (3 µM), or Elexa/Teza (3 µM/10 µM) or ETI (3 µM/10 µM/5 µM) prior to lysis. Right panel: CFTR band C densitometry of the Western blot experiments. Asterisks indicate the statistical significance of the treatments vs. DMSO-treated TT001: ** *p* < 0.01; *** *p* < 0.001.

## Data Availability

All the data are contained within the article or Appendix A.

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
