# Peer review of "The L467F-F508del Complex Allele Hampers Pharmacological Rescue of Mutant CFTR by Elexacaftor/Tezacaftor/Ivacaftor in Cystic Fibrosis Patients: The Value of the Ex Vivo Nasal Epithelial Model to Address Non-Responders to CFTR-Modulating Drugs"

_ijms, 2022, doi:10.3390/ijms23063175_

Round 1

Reviewer 1 Report

In the manuscript by Sondo et al., the authors examined the effect of Trikafta in different cell models (FRT, CFBE, primary nasal epithelial cells) bearing L467F/F508del complex allele. This is a remarkably well-written submission and the data are interesting. The manuscript perfectly fits into the scope of the journal. Hence, this manuscript could be interesting for the readers and I recommend it for publication.

Author Response

REVIEWER 1’S COMMENTS:

In the manuscript by Sondo et al., the authors examined the effect of Trikafta in different cell models (FRT, CFBE, primary nasal epithelial cells) bearing L467F/F508del complex allele. This is a remarkably well-written submission and the data are interesting. The manuscript perfectly fits into the scope of the journal. Hence, this manuscript could be interesting for the readers and I recommend it for publication.

Our response:

We are very grateful to the Reviewer for his/her generous praise.

Reviewer 2 Report

The manuscript highlight a clinical very important differential diagnosis for non-responding to a highly effective CFTR modulator therapy. Complex alleles are rare, but although underestimated. There effect is almost not well understood and the effect of CFTRmodulators has not been investigated for all combinations. The recommended approach would offer a clinical/practical solution for these individual patients.

Comments:

It has to been mentioned that some complex alleles has been responsive based on in vitro data according to https://www.accessdata.fda.gov/drugsatfda_docs/label/2021/212273s004lbl.pdf( (accessed 08 March 2022).

The recent publication from Raraigh in JCF on complete CFTR gene sequencing in 5058 individuals with cystic fibrosis reported L467L in 15 alleles of genes contains F508del. But these reports are restricted to cohort studies, mainly from US.

Author Response

REVIEWER 2’S COMMENTS:

The manuscript highlight a clinical very important differential diagnosis for non-responding to a highly effective CFTR modulator therapy. Complex alleles are rare, but although underestimated. There effect is almost not well understood and the effect of CFTR modulators has not been investigated for all combinations. The recommended approach would offer a clinical/practical solution for these individual patients.

Our response:

We thank the Reviewer for the positive comments and his/her appreciation of our work.

Comments:

It has to be mentioned that some complex alleles has been responsive based on in vitro data according to https://www.accessdata.fda.gov/drugsatfda_docs/label/2021/212273s004lbl.pdf( (accessed 08 March 2022).

Our response:

We agree with the Reviewer that some complex alleles display responsiveness to CFTR modulators, this point is briefly mentioned in the discussion (lines 324-325). Further studies will be needed to investigate systematically the efficacy of CFTR-modulating drugs on complex alleles. 

The recent publication from Raraigh in JCF on complete CFTR gene sequencing in 5058 individuals with cystic fibrosis reported L467L in 15 alleles of genes contains F508del. But these reports are restricted to cohort studies, mainly from US.

Our response:

We thank the Reviewer for addressing this point. We agree that variants distribution and frequency may be different depending on the genetic structure of the considered population or cohort as in the case of the indicated work by Raraigh et al in which individuals are mainly from US. This will have implications also on the frequency of specific complex alleles which are known to be particularly frequent in certain regional cohorts, as in the case of the A238V-F508del allele in the South of Italy (https://doi.org/10.1038/jhg.2016.15) or the F508del-I1027T allele in Brittany, France (https://doi.org/10.1016/j.jcf.2007.07.009).

Reviewer 3 Report

This is a very important paper that helps in explanation of the mechanisms of the lack of response to the targeted therapies in the new era of CFTR modulators. Congratulations to the authors. 

I have only two minor comments.

  1. Page 5 in the middle of the text - "Disappointingly, we could detect any increase in the extent of CFTR-mediated current neither in epithelia treated with correctors Elexa/Teza alone nor in epithelia treated with ETI (Figure 1A,B).  Isn't it - Disappointingly, we could not detect any increase in the extent of CFTR-mediated....
  2. Page 9. In the last sentance  " Epithelia were treated for 24 h with
    vehicle alone (DMSO) or Elexa (3 µM), or Elexa/Teza (3 µM / 10 µM) or ETI (3 10 µM / 10 µM / 5 µM)"   - is there some mistake?

Author Response

REVIEWER 3’S COMMENTS:

This is a very important paper that helps in explanation of the mechanisms of the lack of response to the targeted therapies in the new era of CFTR modulators. Congratulations to the authors. 

Our response:

We thank the Reviewer for his/her appraisal of our work.

I have only two minor comments.

  1. Page 5 in the middle of the text - "Disappointingly, we could detect any increase in the extent of CFTR-mediated current neither in epithelia treated with correctors Elexa/Teza alone nor in epithelia treated with ETI (Figure 1A,B). Isn't it - Disappointingly, we could not detect any increase in the extent of CFTR-mediated....

Our response:

We thank the Reviewer for the revision of this sentence. We have modified accordingly the text.

  1. Page 9. In the last sentance  " Epithelia were treated for 24 h with vehicle alone (DMSO) or Elexa (3 µM), or Elexa/Teza (3 µM / 10 µM) or ETI (3 10 µM / 10 µM / 5 µM)"   - is there some mistake?

Our response:

We thank the Reviewer for highlighting this mistake. We have modified the text, that should read: ETI (3 µM / 10 µM / 5 µM).